# Structural Studies of Mexican Husk Tomato (*Physalis ixocarpa*) Fruit Cutin

**DOI:** 10.3390/molecules29010184

**Published:** 2023-12-28

**Authors:** Daniel Arrieta-Baez, Camila Quezada Huerta, Giovana Simone Rojas-Torres, María de Jesús Perea-Flores, Héctor Francisco Mendoza-León, Mayra Beatriz Gómez-Patiño

**Affiliations:** 1Instituto Politécnico Nacional—(Centro de Nanociencias y Micro y Nanotecnologías), Unidad Profesional Adolfo López Mateos, Col. Zacatenco, Mexico City 07738, Mexico; darrieta@ipn.mx (D.A.-B.); mpereaf@ipn.mx (M.d.J.P.-F.); hfmendoza@ipn.mx (H.F.M.-L.); 2Instituto Politécnico Nacional, Escuela Superior de Ingeniería Química e Industrias Extractivas, Unidad Profesional Adolfo López Mateos, Av. Luis Enrique Erro S/N, Colonia Lindavista 07738, Mexico; camila.qhstar@gmail.com (C.Q.H.); giiovanasiimone@gmail.com (G.S.R.-T.)

**Keywords:** husk tomato, cutin, CPMAS ^13^C NMR, UHPLC-MS, CLSM, SEM

## Abstract

Green tomato (*Physalis ixocarpa*) is a specie native to Mexico, and it is known as “tomatillo” or “husk tomato”. The fruit contains vitamins, minerals, phenolic compounds, and steroidal lactones, presenting antimicrobial activity and antinarcotic effects. Therefore, it is not only used in traditional Mexican cuisine, but also in traditional medicine to relieve some discomforts such as fever, cough, and amygdalitis. However, it is a perishable fruit whose shelf life is very short. As a part of the peel, cuticle, and epicuticular waxes represent the most important part in plant protection, and the specific composition and structural characterization are significant to know how this protective biopolymer keeps quality characteristics in fresh fruits. *P. ixocarpa* cutin was obtained by enzymatic treatments (cellulase, hemicellulose, and pectinase) and different concentrations of TFA, and studied through Cross Polarization Magic Angle Spinning Nuclear Magnetic Resonance (CPMAS ^13^C NMR), Ultra-High Performance Liquid Chromatography coupled to Mass Spectrometry (UHPLC-MS), and was morphologically characterized by Confocal Laser Scanning Microscopy (CLSM) and Scanning Electron Microscopy (SEM). The main constituents identified under the basis of UHPLC-MS analysis were 9,10,18-trihydroxy-octadecanoic acid and 9,10-epoxy-18-hydroxy-octadecanoic acid with 44.7 and 37.5%, respectively. The C16 absence and low occurrence of phenolic compounds, besides the presence of glandular trichomes, which do not allow a continuous layer on the surface of the fruit, could be related to a lower shelf life compared with other common fruits such as tomato (*Solanum lycopersicum*).

## 1. Introduction

Green tomato (*Physalis ixocarpa*) is a species native to Mexico and part of Central America, although it currently grows in tropical and subtropical regions worldwide. It belongs to the Solanaceae family, and is known as “tomatillo” or “husk tomato” [1,2]. Its production in Mexico represents 4.25% of the total vegetable area, and the main producing states are Sinaloa, Zacatecas, Michoacán, Puebla, Jalisco, Tlaxcala, and Morelos [3]. Husk tomato has been widely used in Mexican gastronomy since prehispanic times. In fact, there are reports of the recipes from Maya and Aztec cultures [4]. The fruit contains vitamins, minerals, phenolic compounds, and steroidal lactones, presenting antimicrobial activity and antinarcotic effects [5,6,7]. Therefore, it is not only used in traditional cuisine in Mexico, but also in traditional medicine to relieve some symptoms such as fever, cough, and amygdalitis. However, it is a perishable fruit whose shelf life is very short, perhaps related to the type of cuticle it has [8]. Like many other fruits, husk tomato has a hydrophobic cuticle located in the primary tissues’ outermost extracellular matrix of the epidermis (Figure 1).

The cuticle is an external protection that plants have against the surrounding environment, and we find it covering the outermost cell walls of the aerial parts of the plant. The main role is to protect the plant against uncontrolled water loss, besides physical stabilization and pathogen infection reduction, is that it offers resistance against herbivory, and is involved in several plant development processes [9,10,11,12]. This extracellular membrane is mainly formed by polymerized lipids, whose structure and composition vary according to the type of plant, organs, and growth stage. It is formed by a matrix of cutin and a wax layer [13,14,15]. Cutin is an insoluble polymer conformed by long-chain fatty acids that are hydroxylated and, in some cases, also epoxylated [16,17,18,19].

Studies on the monomeric composition of fruit cutin have been carried out mainly in tomato (*Solanum lycopersicum*) [20,21,22], lime (*Citrus aurantifolia*), and grapefruit (*Citrus paradisi*) [23,24], green pepper (*Capsicum annum*), pumpkin (*Cucurbita pepo*) [20], cucumber (*Cucumis sativus*) [25]; berries: bilberry (*Vaccinium myrtillus*), black currant (*Ribes nigrum*) cranberry (*Vaccinium oxycoccos*), lingonberry (*Vaccinium vitis-idaea*), sea buckthorn (*Hippophaë rhamnoides*), black chokeberry (*Aronia melanocarpa*), cloudberry (*Rubus chamaemorus*), crowberry (*Empetrum nigrum*), raspberry (*Rubus idaeus*), rosehip (*Rosa rugosa*), rowanberry (*Sorbus aucuparia*), strawberry (*Fragaria × ananassa*) [26,27]; and Golden Delicious and Red Delicious apples (*Malus domestica*) [28]. The monomeric composition is mostly of long-chain compounds of 16 and 18 atoms of carbon, which are in a complex, three-dimensional network linked by primary and secondary ester bonds [17,18]. This monomeric composition of cutin has been obtained by depolymerization with alkaline hydrolysis KOH-MeOH 1.5 M [29,30]. Most of the monomers reported under these conditions belong to the C16 long-chain fatty acids family, where the major components are 10,16-dihydroxyhexadecanoic and/or 9,16-dihydroxyhexadecanoic acids [20,31]. However, C18 monomers such as 9,10-epoxy-18-hydroxyoctadecanoic and 9,10,18-trihydroxyoctadecanoic acids have been found as a major component in other fruit cutins [20,26,27].

As has been told, cutin is mainly composed of hydroxy- and epoxy hydroxy C16 and C18 fatty acids and glycerol, but plants species could have different cutin composition, especially when they are in developing stages [11,32], and some environmental stresses [33]. The properties of the cuticle can be affected by the variation in its monomeric composition, affecting its structure. For example, these changes have been correlated to the resistance of some plants to *Erysiphe polygoni* [34] and *Botrytis cinerea* [35]. The presence of some monomers that contain specific groups, such as epoxy groups, may imply the union of some chemical substances, in the same way the mechanical properties of the cuticle have been related to the monomeric composition of this [36,37]. It has been shown that the presence of hydroxyl groups increases its hydrophilic character, observing greater elasticity [38], and thereby affects cross-linking through polyester bonds [39].

Polysaccharides are another important domain in the cuticular matrix, and could be responsible for the low yield in the hydrolysis reactions. For non-cellulosic polysaccharides, trifluoroacetic acid (TFA) is also an option to remove or hydrolyze them, with the advantage that its volatility allows for removal by evaporation or lyophilization rather than a neutralization step [40]. Depending on the aqueous concentration that is employed, TFA may serve as a solvent or hydrolytic reagent for cellulose and lignocellulosic materials [41,42].

The aim of the present work was to study the monomeric composition of the cutin of husk tomato (*Physalis ixocarpa*) fruit, as well as the morphology changes in the surface of the cuticle due to acid and alkaline hydrolysis, and correlate both studies to understand the role in the low shelf life of this tomato fruit.

## 2. Results and Discussion

### 2.1. Cross Polarization Magic Angle Spinning ^13^C Nuclear Magnetic Resonance (CPMAS ^13^C NMR) of Mexican Husk Tomato Cutin

Cutin from Mexican husk tomato (*Physalis ixocarpa*) was isolated according to some reported enzymatic treatments [43].

After this, cutin was analyzed by CPMAS ^13^C NMR. Most of the signals obtained have been previously reported for other cutins and corroborated in this analysis according to the next assignments (Figure 2A) [44]: bulk methylenes were the main peaks observed at 20–40 ppm, oxygenated aliphatic carbons at 55–85 ppm, aromatics, and olefins at 105–155 ppm and carbonyl groups at 173 ppm. Carbohydrate molecular signals were detected as well at 60 ppm (C6), C2,3,5 70–75 ppm (C2,3,5), 83 ppm (C4), and 105 ppm (C1). In comparison with different analyses done with some other cutins previously reported, peaks at 56 and 64 ppm were observed in Mexican husk tomato cutin. These peaks were assigned to epoxilated long-chain aliphatic acids (See Appendix A).

When 0.01 M TFA was used, these peaks were still present in the sample (Figure 2B). However, when a higher concentration was used (0.1 M and 1.0 M), these peaks were removed (Figure 2C,D). The soluble material removed was subject to ^1^H and ^13^C NMR and Direct Injection Electrospray Mass Spectrometry (DIESI-MS) analysis. In both cases, when alkaline hydrolysis was used with untreated cutin material and 0.01 M TFA insoluble material, cellulosic material was recovered (Figure 2E) (See Appendix A), and the soluble compounds were analyzed by means of UHPLC-MS and DIESI-MS analysis.

### 2.2. Ultra-High Performance Liquid Chromatographic (UHPLC-MS) and Direct Injection Electrospray Mass Spectrometry (DIESI-MS) Analysis of the Physalis ixocarpa Cutin TFA and ALKaline Hydrolysis (KOH/MeOH) Products

When alkaline hydrolysis (KOH/MeOH 1.5 M) was used in the cutin from the enzymatic hydrolysis, around ≈87% of the cuticular material was hydrolyzed and the monomeric components were obtained.

Soluble products from the alkaline hydrolysis of the cuticular material were analyzed by means of UHPLC-ESI analysis in negative mode (Figure 3), and the compounds identified by the *ms/ms* analysis are reported in Table 1.

From the alkaline hydrolysis of the intact *P. ixocarpa* cutin, the main constituents identified under the basis of UHPLC-MS analysis were 9,10,18-trihydroxy-octadecanoic acid and 9,10-epoxy-18-hydroxy-octadecanoic acid with 44.7 and 37.5%, respectively (Table 1). 10,16-Dihydroxy Palmitic Acid has been identified in different cutins such as citrus, green pepper, and tomato, as one of the most important C16 alkanoic acids [20]. However, this monomer was not identified in the *P. ixocarpa* cutin, only the palmitic acid, which could be one of its precursors. Epoxylated compounds identified in cutins, along with 9,10,18-trihydroxy-octadecanoic acid and 9,10,18-trihydroxy-octadecanoic acid, could be derived after subsequent oxidation reactions of linoleic acid, detected in the cutin as well. The predominance of C16 long-chain acids in cutins is common and agrees with previous cutin reports [20]. However, it is important to highlight that most of the main monomers (98%) in the Mexican husk tomato cutin are C18 long-chain acids. Aromatic compounds and carbohydrates were not detected by MS analysis in *P. ixocarpa* cutin, maybe because of their low presence and/or their higher polarity, which agrees with the NMR analysis.

When the cuticular material of *P. ixocarpa* was subject to 0.01 M TFA hydrolysis, there were no monomeric components removed. However, for higher concentrations of TFA (0.1 M and 1.0 M) only 9,10-epoxy-18-hydroxy-octadecanoic acid was recovered (See Appendix A). This compound was identified according to DIESI-MS analysis of the soluble compounds (Figure 4A) and agrees with those signals previously observed in CPMAS ^13^C NMR.

After the 9,10-epoxy-18-hydroxy-octadecanoic acid was removed, cuticular material was hydrolyzed using KOH/MeOH 1.5 M. According to the DIESI-MS analysis of the hydrolyzed components, all the previous monomers were identified except for 9,10-epoxy-18-hydroxy-octadecanoic acid (Figure 4B). This would indicate that this monomer is located on the surface of the cutin and is more exposed to hydrolysis. However, it is not present in what could be considered the main network of this important bioprotector polyester.

### 2.3. Confocal Laser Scanning Microscopy (CLSM) of Mexican Husk Tomato (Physalis ixocarpa) Cutin

The study of cutins has represented a challenge for its analysis because it is a material that is very resistant to different hydrolysis treatments, and that is why CLSM has become a valuable tool to analyze the autofluorescence of cutins of plant tissues, since previous treatments or physical sectioning of the sample are not required, so the cutin samples remain unchanged [45]. Cutin from *P. ixocarpa* and the insoluble material obtained after the TFA treatments were analyzed by CLSM without artificial staining or physical sectioning, and most of the cellular structures could be observed because of their autofluorescence. As seen in Figure 4, the micrographs of the internal wall of the samples treated with 0.01 M and 0.1 M show a small variation in the thickness and height of the anticlinal walls with respect to the untreated sample (Figure 5A–C). According to the results obtained by Mass Spectrometry, the epoxylated components are those that were eliminated with the 0.1 M TFA treatment, which would indicate that the structure of the cuticular material obtained from the treatment with 0.1 M TFA would only be hydroxylated long-chain aliphatics. This agrees with the fact that the network is formed by polyesters cross-linked with primary and secondary hydroxyls, where the epoxylated components could not present this cross-linking. However, the images obtained from the cuticular residues after treatment with 1.0 M TFA showed differences compared to those without treatment (Figure 5D). In this case, we can observe that the defined edges disappear, leaving only a few traces, which could be related to the removal of components associated with cellular structures. Relating these results with those of the solid-state NMR, the loss of polysaccharides (Figure 2C,D) can reduce the elasticity of the anticlinal walls, causing irregular cell walls to form (Figure 5C,D).

The same results were observed in the external face; well-defined geometric figures are still observed, and when 1.0 M TFA treatment was used, most of the defined edges disappeared. However, in this case, some points were observed indicating the presence of trichomes (white arrows in Figure 5E–G), which are fine outgrowths or appendages on plants, founded in the plant surfaces protecting, amongst others, against pathogen and herbivore attacks, being commonly found in solanaceous plant species.

The insoluble cuticular material obtained after the alkaline hydrolysis (KOH/MeOH 1.5 M) was analyzed by solid-state NMR and CLSM and the results indicate only the cellulosic material (See Appendix A).

### 2.4. Scanning Electron Microscopy (SEM) Analysis of Mexican Husk Tomato (Physalis ixocarpa) Cutin

In order to have the best record in the image analysis, the sample used for CLSM was used in the SEM analysis, so it was prepared according to the sputtering coating procedure, and the images were captured by varying parameters such as acceleration voltage, working distance and magnifications.

According to Figure 6, both faces look different than those observed with CLSM. For the inner face analysis, in Figure 6A, irregular geometric figures formed by ribs are observed for the cutin without any treatment. As the TFA is used in treatments of 0.01 M, 0.1 M and 1.0 M changes the well-delimited figures, and the roughness of the deep zones can be related to the effect of the TFA on the non-cellulosic sugars remaining after the enzymatic treatments with pectinase, cellulase, and hemicellulase. The walls forming the irregular figures are probably formed by the remaining non-cellulosic sugars, and upon being subjected to TFA treatment, these compounds are released, leaving a structure formed mainly of aliphatic components (0.01 M TFA treatment) according to solid-state NMR analysis (Figure 2B).

Images obtained from the cuticular residues after the treatment with 0.1 M TFA, show still irregular figures, although the average height of the ribs is lower (5.66 nm), compared with those without any treatment (6.87 nm) (Figure 7). This observation is related to CPMAS ^13^C NMR analysis previously described, where no changes in the signals were observed before and after the treatment. It seems that polysaccharides, which remain in equal percent of aliphatic compounds, gave support to the cutin structure in order not to lose its shape. This is not observed in the cutin, where most of the polysaccharides were removed after the 1.0 M TFA treatment, where the anticlinal walls of the cutin lost its shape (Figure 6D). The TFA in *P. ixocarpa* cutin seems to dissolve some non-cellulosic polysaccharides and hydrolyze or release aliphatic compounds in a relationship that is constant in the surface of the cuticle, which causes the walls to decrease in height and thickness, but keeps the same percent of components in the remaining blocks.

For the outer face, only the marks for the inner ribs are observed in a non-planar cutin surface. As we can see in Figure 6F,G, the surface is being “removed” after treatment with TFA, until some ribs appear after the 1.0 M TFA treatment.

Besides the biochemical changes throughout the growth process of this fruit, the mechanical properties of the cutin could be related to the chemical composition affecting the viscoelastic behavior of the cuticle. Studies of tomato (*S. lycopersicum*) cutin have demonstrated that quantitative changes in its components influence the elastic/viscoelastic behavior of the cuticle [46]. Especially polysaccharides and some phenolic compounds like flavonoids, which are related to the rigidity of the cutin network acting as biochemical modulators [46]. The low presence of polysaccharides and phenolics, related to other fruit cutins, may affect the elasticity of this cutin, initiating a break in the polyesters and causing some enzymatic reactions that could reduce the shelf life of this fruit.

Glandular trichomes observed in the outer face (Figure 8) have been previously reported for *P. ixocarpa*. These trichomes were well defined until 0.01 M TFA treatment and completely disappeared with 1.0 M TFA treatment. More studies are needed to elucidate the role of these trichomes in the shelf life compared with those that do not have them, such as tomato (*Solanum lycopersicum*).

## 3. Materials and Methods

### 3.1. Chemicals

The enzymes *Aspergillus niger* pectinase (EC 3.2.1.15), *A. niger* cellulase (EC 3.2.1.4), and *A. niger* hemicellulose (EC 3.2.1.4) were purchased from Sigma-Aldrich (St Louis, MO, USA). Methanol, water, acetonitrile, and trifluoroacetic acid were purchased from Sigma-Aldrich (St. Louis, MO, USA). Other laboratory chemicals were all of reagent grade or better.

### 3.2. Isolation of Cutin

Ripe Mexican husk tomato fruits (*Physalis ixocarpa*) were bought at a local marketplace in Mexico City, during the autumn–winter 2022 cycle. Cutins were obtained by the methodology described by Arrieta-Baez, D. and Stark, R.E (2006) [42], with some modifications. The fruits without the calyx were placed in boiling water for 30 min in order to eliminate the pulp and seeds, then the resulting pericarps were treated with a solution of sodium acetate 50 mM, pH 4 mixed with pectinase from *Aspergillus niger* (10 mL/L equivalent to 2500 units of pectinase) and stirred for three days at 150 rpm at 44 °C. Afterwards, the cuticle material was washed with water and treated with a solution of sodium acetate 50 mM, pH 5 with cellulase (1 g/L equivalent to 1200 units of cellulase) and hemicellulose (1 g/L equivalent to 1500 units of hemicellulose) from *A. niger* for three days at 150 rpm at 37 °C. Finally, the material was washed with water and filtered, followed by a second wash with methylene chloride:methanol (1:1, *v*/*v*) for 2 h.

### 3.3. Treatment of Cutin with Trifluoroacetic Acid [42]

In total, 150 mg of the obtained cutin was stirred at 115 ± 5 °C in a stoppered flask in separate experiments, using aqueous 0.01 M, 0.1 M, and 1.0 M TFA for 2 h. Each reaction mixture was filtered, and the insoluble material was washed with stirring using chloroform–methanol (1:1 *v*/*v*) for 1 h to give TFA-hydrolyzed cutin. The TFA hydrolyzed cutin was separated by filtration, dried, and analyzed by CPMAS ^13^C NMR. The TFA solution was co-evaporated with methanol to dryness, and the resulting solids were later analyzed by DIESI-MS (−) and solution-state NMR. Each treatment was replicated three times with similar yields and spectroscopic data; subsequent exhaustive treatments yielded no additional soluble products.

### 3.4. Alkaline Hydrolysis of CUTIN and 0.1 M TFA Cutin with KOH [23]

A total of 50 mg of cutin and 0.1 M TFA treated cutin were separately added to a 50 mL of 1.5 M methanolic KOH solution; the mixture was stirred at room temperature for 24 h. After this time, the reaction was filtered, neutralized and monomers were extracted with CHCI_3_-MeOH. The dried extract was weighed, dissolved in CHCI_3_-MeOH and analyzed by DIESI-MS (−). The unreacted material (KOH-hydrolyzed cutin) was analyzed by CPMAS ^13^C NMR.

### 3.5. CPMAS ^13^C NMR of Mexican Husk Tomato Cutin Analysis

Solid-state NMR experiments were carried out on a Varian Instruments Unityplus 300 wide bore spectrometer (Palo Alto, CA, USA) operating at 74.443 MHz of resonance frequency with a customary acquisition time of 30 ms, a delay time of 2 s between successive acquisitions, and a Cross-Polarization (CP) contact time of 1.5 ms. Typically, each 30 mg sample was packed into a 5 mm rotor and supersonic Magical Angle Spinning (MAS) probe from Doty Scientific (Columbia, SC, USA), then spun at 6.00 (0.1 kHz and room temperature for approximately 10 h. The resulting data were processed with 50 Hz of exponential line broadening.

### 3.6. Ultra-High Performance Liquid Chromatography Mass Spectrometry (UHPLC-MS) and Direct Injection Electrospray Mass Spectrometry (DIESI-MS) Analysis

An Ultimate 3000 ultra-performance liquid chromatography (UHPLC) system (Dionexcorp., Sunnyvale, CA, USA) with photo diode array detection (PAD), was coupled to a Bruker MicrOTOF-QII system by an Electrospray Ionization (ESI) interface (BrukerDaltonics, Billerica, MA, USA) for chromatographic and Mass Spectrometry (MS) analysis. For chromatographic separation, a Hypersil C18 column (3.0 μm, 125 × 4.0 mm) (Varian) was used. The mobile phase consisted of 0.1% formic acid in water (A) and acetonitrile (B) using an isocratic program of 5–95% (B) in 5 min. The solvent flow rate was 0.5 mL/min, the column temperature was set to 30 °C, and the detection wavelength was 254 nm. The conditions of MS analysis in the negative ion mode were as follows: drying gas (nitrogen), flow rate, 8 L/min; gas temperature,180 °C; scan range, 50–3000 m/z; end plate off set voltage, −500 V; capillary voltage, 4500 V; nebulizer pressure, 2.5 bar.

The accurate mass data of the molecular ions were processed through the software DataAnalysis 4.0 (Bruker Daltonics Technical Note 008, 2004). The widely accepted accuracy threshold for confirmation of elemental compositions was established at 5 ppm.

DIESI-MS analysis was done on a Bruker MicrOTOF-QII system, using an electrospray ionization (ESI) interface (Bruker Daltonics, Biellerica, MA, USA) operated in the negative ion mode. A solution of 10 µL of the sample resuspended in 1 mL of methanol was filtered with a 0.25 µm polytetrafluoroethylene (PTFE) filter and diluted 1:100 with methanol. Diluted samples were directly infused into the ESI source and analyzed in negative mode. Nitrogen was used with a flow rate of 4 L/min (0.4 Bar) as a drying and nebulizer gas, with a gas temperature of 180 °C and a capillary voltage set to −4500 V. The spectrometer was calibrated with an ESI-TOF tuning mix calibrant (Sigma-Aldrich, Toluca, Estado de México, México).

MS/MS analysis was performed using positive and negative electrospray ionization (ESI+/−), and the obtained fragments were analyzed by a Bruker Compass Data Analysis 4.0 (Bruker Daltonics, Technical Note 008, 2004).

### 3.7. Confocal Laser Scanning Microscopy Analysis

For CLSM analysis, each sample was mounted on glass slices and observed under CLSM (LSM 710 NLO, Carl Zeiss, Jena, Germany) with objective EC Plan-Neofluar 10×/0.3. The laser wavelength excitation was 405, 488, 561 y 633 nm, simultaneously. This capture mode used was a spectral imaging technique that automatically outputs separated channels of multiple labeled samples. This tool detects the autofluorescence signal of cuticular material and some of them were compared experimentally with patrons (cellulose) between 420 to 720 nm. The z-stack images (3D images) were captured by means of the software ZEN 2010 (Carl Zeiss, Germany), at 512 × 512 pixels in RGB color and stored in TFF format at 8 bits.

### 3.8. Scanning Electron Microscopy Analysis

Isolated cuticular membranes of *P. ixocarpa* were observed with a scanning electron microscope (JSM 7800F, JEOL, Akishima, Japan). Cuticular membranes were placed on aluminum holders with a double-sided adhesive carbon film (Plannet Plano, Plano, TX, USA), 1 min coated with carbon (60:40) at 25 mA using an SPI sputter coater and examined in a field emission at 1.5 kV. The sputter conditions, depositing carbon coat thickness of approximately 10 nm, were optimized for the acceleration voltage in the scanning electron microscope.

## 4. Conclusions

Cutin from Mexican husk tomato (*P. ixocarpa*) was obtained and analyzed. This important protective biopolyester mainly showed aliphatic and polysaccharide domains according to the CPMAS ^13^C NMR. According to these analyses, a very small domain of polysaccharides was detected, with no presence of phenolic compounds. Main monomers characterized as 9,10,18-trihydroxy-octadecanoic acid and 9,10-epoxy-18-hydroxy-octadecanoic acid with 44.7 and 37.5%, respectively, were identified by UHPLC-MS in *P. ixocarpa* cutin, with no presence of monosaccharides or phenolic compounds, which agree with solid-state NMR analysis. The main presence of C18 long-chain aliphatic acids with epoxilated compounds, the low presence of polysaccharides and phenolic compounds in the cutin could make it susceptible to a faster maturation process. However, more studies are necessary to understand the physicochemical characteristics at this level and correlate it with the low shelf life, which results in economic losses.

## Figures and Tables

**Figure 1 molecules-29-00184-f001:**
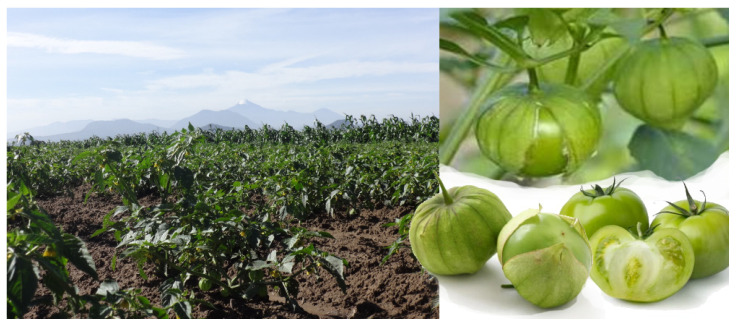
Mexican field of husk tomato (*Physalis ixocarpa*) and fruits with and without the “husk”.

**Figure 2 molecules-29-00184-f002:**
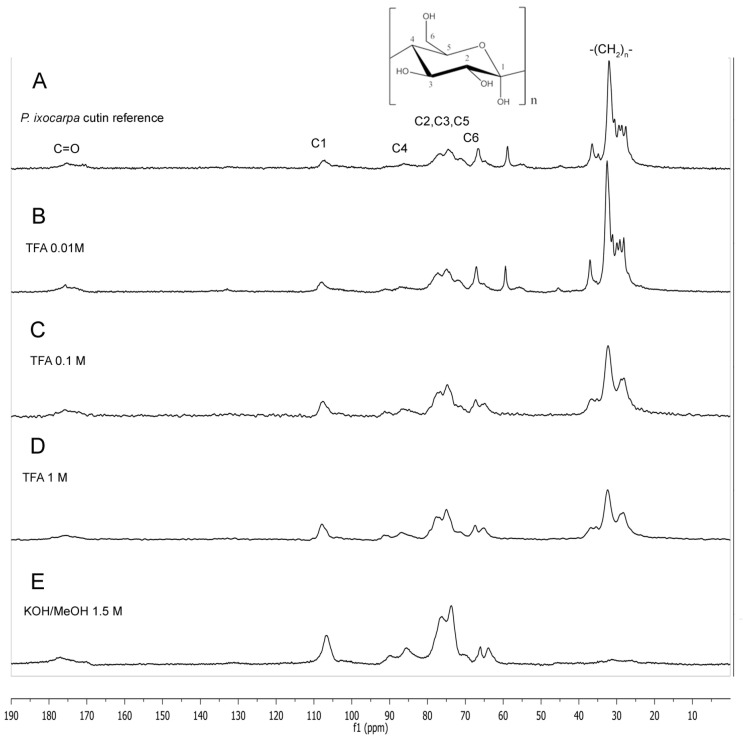
CPMAS ^13^C NMR spectra of Mexican husk tomato (*P. ixocarpa*) cutin. (**A**) Cutin after enzymatic treatment (reference). (**B**–**D**) Insoluble cuticular material after different TFA treatments (0.01, 0.1 and 1.0 M), and (**E**) insoluble cuticular material after alkaline hydrolysis treatments. Besides the aliphatic and carbohydrate peaks assigned, epoxylated compounds were detected because of their signals at δ 56 and 64 ppm.

**Figure 3 molecules-29-00184-f003:**
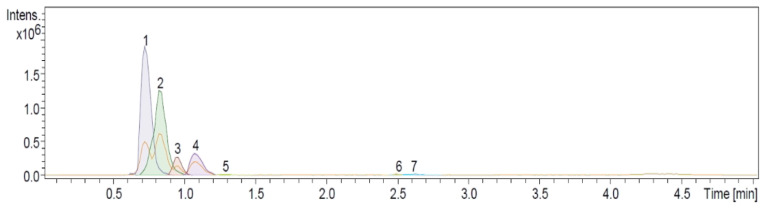
Dissect analysis of the UHPLC-MS chromatogram of soluble products from alkaline hydrolysis (KOH/MeOH, 1.5 M) of Mexican husk tomato (*P. ixocarpa*) cutin.

**Figure 4 molecules-29-00184-f004:**
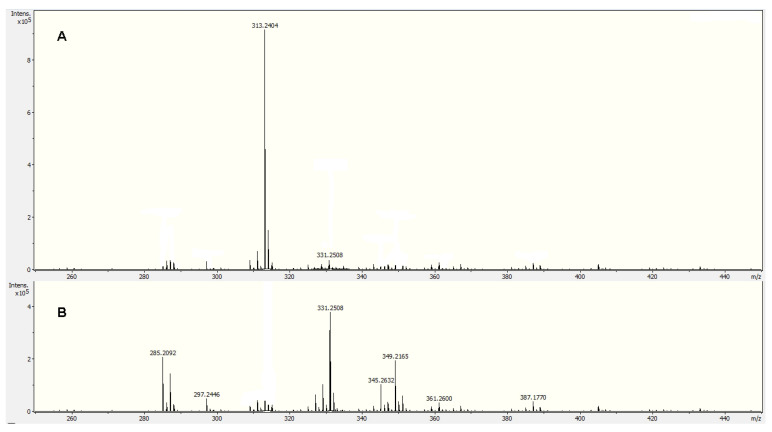
DIESI-MS chromatogram of cutin from Mexican husk tomato (*P. ixocarpa*). (**A**) Soluble compounds of the 100 mM TFA hydrolysis. (**B**) Soluble compounds of the alkaline hydrolysis of the cuticular material after of 100 mM TFA.

**Figure 5 molecules-29-00184-f005:**
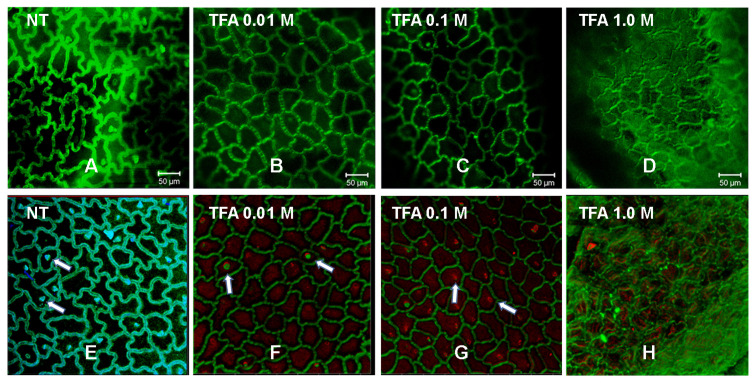
CLSM micrographs of cutin from *P. ixocarpa*. Inner face (**A**–**D**), and outer face (**E**–**H**). NT: No treatment, white arrows: glandular trichomes.

**Figure 6 molecules-29-00184-f006:**
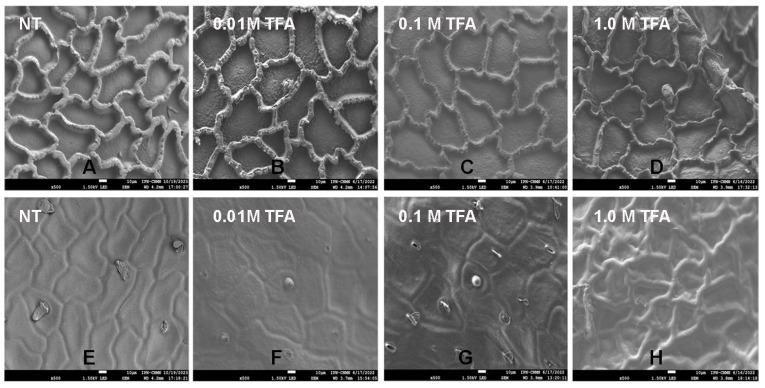
SEM micrographs of cutins from Mexican husk tomato (*P. ixocarpa*). Inner face (**A**–**D**), and outer face (**E**–**H**).

**Figure 7 molecules-29-00184-f007:**
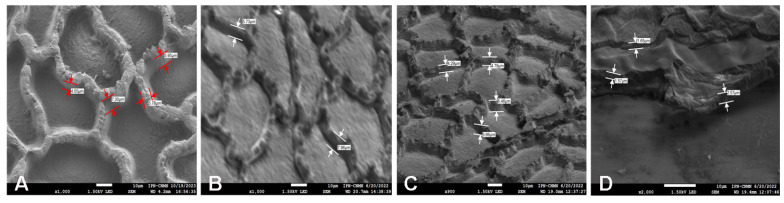
Measure of the ribs (**A**) normal cutin of P. ixocarpa (**B**) 0.01 M TFA (**C**): 0.1 M TFA and (**D**) 1.0 M TFA.

**Figure 8 molecules-29-00184-f008:**
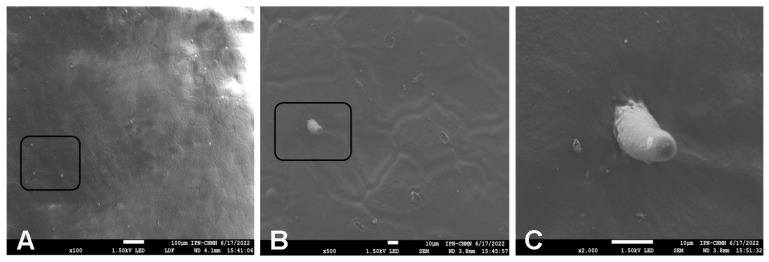
SEM micrographs of glandular trichomes in Mexican husk tomato (*P. ixocarpa*) cutin at different scales: (**A**) 100 µm (**B**) 20 µm (**C**) 10 µm.

**Table 1 molecules-29-00184-t001:** Identification of the main compounds in the soluble fraction of the alkaline hydrolysis of the *P. ixocarpa* cutin.

#	Rt	Name	Fragments	[M-H]^−^_obs_	[M-H]^−^_exact_	Formula	Error	% RA
1	0.7	9,10,18-trihydroxy-octadecanoic acid	172, 189, 202, 287	331.2526	331.2490	C_18_H_36_O_5_	−10.7	44.7
2	0.8	9,10-epoxy-18-hydroxy-octadecanoic acid	171, 184, 269	313.2424	313.2384	C_18_H_34_O_4_	−12.7	37.5
3	0.9	9,10-epoxy-12Z-18-hydroxy-octadecenoic acid	142, 184, 210, 267	311.2267	311.2216	C_18_H_32_O_4_	−12.5	5.7
4	1.1	18-hydroxyoleic acid	155, 253	297.2435	297.2424	C_18_H_34_O_3_	12.1	9.7
5	1.3	18-hydroxyoctadecanoic acid	255, 282	299.2630	299.2580	C_18_H_36_O_3_	12.8	0.6
6	2.5	Oleic acid	142, 237	281.2511	281.2475	C_18_H_34_O_2_	−9.0	0.8
7	2.6	Palmitic acid	211	255.2326	255.2330	C_16_H_32_O_2_	−0.8	0.8

[M-H]**^−^**_exact_: molecular weight exact, [M-H]**^−^**_obs_: molecular weight observed, % RA: % relative area. Error [ppm]: absolute value of the deviation between measured mass and theoretical mass of the selected peak in [ppm].

## Data Availability

Data are contained within the article and Appendix A.

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
