# Peer review of "Structural Studies of Mexican Husk Tomato (Physalis ixocarpa) Fruit Cutin"

_molecules, 2023, doi:10.3390/molecules29010184_

Round 1

Reviewer 1 Report

Comments and Suggestions for Authors

The paper entitled “STRUCTURAL STUDIES OF MEXICAN HUSK TOMATO (Physalis ixocarpa) FRUIT CUTIN” by Arrieta-Baez et al. is an interesting study where the cutin of “tomatillo” is characterized by different chemical and microscopic techniques. The topic deserves attention and the manuscript is, in general, well-written and presents robust experimental info.

However, I have some minor comments that can help to improve the overall quality of the work:

1.- A photo of the “tomatillo” with the husk should be provided.

2.- l3. “Physalis ixocarpa” should be in italic.

3.- In the abstract the different methodologies used to isolate the cutin (enzymatic treatment, TFA, KOH…) should be summarized.

4.- Legend of Figure 1. Please, describe what is each letter (in particular, B, C, D, E). I am confusing and I am not sure if the NMR of the cutin or the removed fraction are showed.

5.- l157-162. Regarding the disappearance of 9,10-epoxy-18-hydroxy-octadecanoic acid, are the authors considered the opening of the epoxide during the different chemical treatments with TFA, KOH… Such an opening would exactly form 9,10,18-trihydroxy-octadecanoic acid (the other main monomer). It is just a suggestion (I had the same problem with the cutin from pepper fruit).

6.- Finally, about the presence of polysaccharides in the cutin after the enzymatic treatment. Maybe, to consider cutin a mixture of polyester and polysaccharides is not very right. I think the authors should describe how polysaccharides and cutin (the polyester fraction of cuticle) are associated in other plants. Maybe, in this case, they are intimately blended (differently to tomato fruits) and their separation is more complicated.

Comments on the Quality of English Language

There minor errors in the text.

Author Response

Thank you for your comments.

Reviewer 2 Report

Comments and Suggestions for Authors

The main problem of the manuscript "Structural studies of mexican husk tomato (Physalis ixocarpa) fruit cutin" is that this title emphasizes the presence of structural studies that are not actually mentioned in the text. Although the experimental data are interesting, they only concern the chemical characterization of the cutin of the Mexican tomato. The identification of the chemical compounds does not constitute a structural analysis. The arguments made are correct, but they do not address the main question stated in the title. For this reason, the authors need to add information obtained with specific chemical approaches that provide structural data:

1) spectroscopic measurements (e.g. FT-IR)

2) surface spectrometric measurements (e.g. ToF-SIMS, XPS) or X-ray spectrometer (XRF) with the aim of recognizing the chemical groups present in the system and evaluating possible structural interactions.

3) Solid-state NMR spectroscopy. To investigate the structure present in the plant environment and to detect the interaction between different structures or macromolecules. Figure 3 is in need of improvement (poor quality).

Supplementary materials:

Fig S1 and S2 are not necessary.

Comments on the Quality of English Language

I suggest a general check of the English style in order to refine the paper style.

Author Response

Thank you for your comments.

Reviewer 3 Report

Comments and Suggestions for Authors

The ideas and the results are clearly presented in the manuscript, So, it requires minor revision on the bases of following comments.

Figures and Tables:

1.       Please improve quality of Figure 1, 2 and 3. These figures look blurred. Good graphics attract lot of the researchers to read the research work.

2.       Please check bibliography. Few name of journals are abbreviated while others are not.

3.       Please remove grammatical mistakes.  

4.       Cite some latest articles on this specific problem in introduction section. Latest articles mean, published in 2023.

5.       Move Figure S6 from supplementary file to main manuscript.

6.       Compare NMR with related literature.

Others that updated by reviewer:

1.       The most prime motive of the research should be more elaborated in introduction section with some latest references.

2.       References are appropriate but in some references, name of journal is abbreviated whereas in others, abbreviations are not used. For example, see reference numbers 8, 32, 38.

3.       The conclusion is fine but it should be concise.

4.       Please mention how the research reported in the manuscript is more fruitful with respect to the similar research reported in literature.

5.       Material and method section: please mention why not fruits without the calyx are placed in boiling water for 2 hours or more?

6.       Please add full name of abbreviations when used for the first time in the manuscript.

7.       Please mention the limitations of present research work in introduction section.

Comments on the Quality of English Language

Please read whole manuscript and remove grammatical mistakes

Author Response

Thank you for your comments.

Reviewer 4 Report

Comments and Suggestions for Authors

The manuscript is covering the compositional profiling and structural characerization of husk tomato cutin using CPMAS 13C NMR, LC-MS, CLSM and SEM. It is interesting to find that the low concentration of phenolic compounds and the presence of glandular trichomes may responsible to the lower shelf life of the fruit. These results enhance our understanding of husk tomato cutin composition and its limited shelf life.

Remarks:

1.    Adding a control sample, such as tomato Solanum lycopersicum, would strengthen the study.

2.    It's unclear why DIESI-MS was used for cutin composition analysis.

3.    Although both positive and negative ionization modes were analyzed in UHPLC-ESI-MS, only the results of the negative mode were presented. Clarification is needed.

4.    Figure 2 requires an explanation of the chromatograms' different colors in the figure caption.

5.    Table 1 should include the fragments of each compound.

6.    Line 140 should explain the complete name of DHPA.

7.    Line 147, should be “most of the main monomers (98%)”.

8.    For Figure 3, a high-resolution chromatogram image would be more appropriate than the current snapshot.

Author Response

Thank you for your comments.

Round 2

Reviewer 2 Report

Comments and Suggestions for Authors

Dear authors,

Your comments are acceptable, but not sufficient to modify my idea. The title of the manuscript should be changed.

However, I accept the revised manuscprit.

Comments on the Quality of English Language

Minor editing of English language required

Reviewer 4 Report

Comments and Suggestions for Authors

The authors have addressed the reviewers' comments.